# Insignificant Difference in Biocompatibility of Regenerated Silk Fibroin Prepared with Ternary Reagent Compared with Regenerated Silk Fibroin Prepared with Lithium Bromide

**DOI:** 10.3390/polym14183903

**Published:** 2022-09-18

**Authors:** Guotao Cheng, Xin Wang, Mengqiu Wu, Siyuan Wu, Lan Cheng, Xiaoning Zhang, Fangyin Dai

**Affiliations:** 1State Key Laboratory of Silkworm Genome Biology, Key Laboratory of Sericultural Biology and Genetic Breeding, Ministry of Agriculture and Rural Affairs, College of Sericulture & Textile and Biomass Sciences, Southwest University, Chongqing 400715, China; 2Center for Joint Surgery, Southwest Hospital, Third Military Medical University (Army Medical University), Chongqing 400038, China

**Keywords:** silk fibroin, dissolution, ternary reagent, lithium bromide, biocompatibility, hemolysis

## Abstract

*Bombyx mori* silk fibroin (SF) is widely used in the field of biomaterials due to its excellent biocompatibility and mechanical properties. However, SF cannot be used directly in many applications and needs to be dissolved first. Lithium bromide (LiBr) is a traditional solvent which is usually used to dissolve SF. However, LiBr has several limitations, e.g., it is expensive, it is toxic to organisms, and it is environmentally unfriendly. Herein, we investigate the possibility of developing a ternary reagent system that is inexpensive, non-toxic to organisms, and environmentally friendly as an alternative for silk fibroin solubilization. The results confirm that regenerated silk fibroin (RSF) prepared using a ternary reagent has the same morphology and amino acid composition as that prepared using LiBr, but the RSF prepared using a ternary reagent still had a small amount of calcium residue even after long-term dialysis. Further research found that the residual calcium does not cause significant differences in the structure and biological performance of the RSF, such as its cytotoxicity, blood compatibility, and antibacterial properties. Therefore, we believe that ternary reagents are an ideal alternative solvent for dissolving SF.

## 1. Introduction

Silk fibroin (SF), a natural protein purified from *Bombyx mori* silkworm cocoons, has been used as a biomaterial in numerous applications, including tissue engineering, drug delivery, and implanted devices. The reasons for its widespread use include the ease with which it can be processed, its excellent biocompatibility, and its tunable mechanical and degradation properties [1,2,3,4]. Natural SF in silkworms is water soluble and behaves like a typical soluble polymer in solution [5,6]. However, when silkworms are spun, the spun fiber is insoluble in water due to protein hydration and structural transition to β-sheets [7]. Therefore, insoluble silk fibers require multiple processing steps before being processed into different forms. The processing of SF predominantly involves degumming, dissolution, dialysis, and forming. Among these, dissolution is an essential step. There are various dissolution systems to dissolve the SF [8,9], such as high-concentration neutral salt solvents [9], strong acids such as sulfuric acid [10] and formic acid [11,12], ionic liquids [13], and composite solvents such as salt-alkali [14], salt-alcohol, and salt-acid [9,15]. More novel solvents that can dissolve SF have also been discovered and reported [16,17]. Different solvents have different solubilities [18] and affect the molecular weight distribution [19], aggregate structure, morphology, and viscosity [9] of the regenerated silk fibroin (RSF).

Among these solvents, lithium bromide (LiBr) and calcium chloride (CaCl_2_) have attracted the attention of researchers. A 9.0–9.3 mol LiBr aqueous solution can swiftly dissolve SF at 70 °C–80 °C [9]. A CaCl_2_ aqueous solution can still dissolve silk fibroin, but the dissolution rate is slower than that of LiBr at the same temperature because Ca^2+^ cannot easily penetrate the crystalline region of the SF. Nevertheless, calcium chloride has attracted the attention of researchers [8,9]. The addition of small-molecule plasticizers can significantly alleviate this problem [20,21,22]. For example, CaCl_2_-methanol can easily break down silk fibroin into nanofibers [20]. The mixture of CaCl_2_, EtOH, and H_2_O, with a molar ratio of 1:2:8, also known as Ajisawa’s reagent or a ternary reagent, can dissolve SF rapidly because EtOH can bring Ca^2+^ into the crystalline region of the SF [21]. We anticipate the use of the ternary reagent in industrial production due to the small amount of inorganic salt required (1 mol CaCl_2_) and its low cost [23].

However, some studies suggest that the two solvents still produce differences in the dissolution and regeneration of SF. For example, RSF prepared with a ternary reagent has a high aggregation rate during the dialysis step. However, RSF dissolved in aqueous LiBr does not form aggregates during dialysis and shows high stability [24]. The variation between them can further influence the final properties of the RSF materials, such as the degree of crystallinity, conformational transitions, thermal stability, and surface structures [21]. More importantly, the reserves of Li on the earth are very low, and this coupled with the demand for it in the new energy industry has meant that the price of lithium salt has become unfeasible. In addition, many studies report that Li is highly toxic to humans and the environment, and that it will cause great environmental pollution if it is directly discharged into the environment without treatment [25,26,27]. The treatment of lithium-containing wastewater increases production costs and corporate burdens, so LiBr is not economically viable for large-scale production of RSF [17]. In contrast, Ca is an essential component of living organisms, has no apparent environmental toxicity, and is inexpensive. Therefore, it is crucial to systematically and extensively study the differences between lithium bromide and ternary solvents to find an inexpensive alternative solvent for the large-scale production of RSF.

For this paper, we prepared two RSFs, one using LiBr (RSF-Li) and one using a ternary reagent (RSF-Ca), and comprehensively investigated their compositions, structures, biocompatibility, antibacterial properties, and applications, with the aim of identifying a cost-effective and environmentally friendly solvent for the large-scale production of RSF. The solubilization and regeneration of SF is a long, multi-step process. Every tiny difference in processing results in a change in the final properties of the RSF [28]. Molecular weight is one of the key parameters [29]. Lower molecular weight leads to difficulty in shaping the fibers during spinning, weaker mechanical properties, and an enhanced degradation rate [30,31,32]. It can be regulated by dissolution conditions, such as temperature and time [33]. To avoid this effect, the RSF samples prepared from the two different solvents in this study, under controlled conditions, had identical molecular weight distributions. Similarly, we selected cocoons of the same variety and production season as the raw material for SF extraction.

## 2. Material and Methods

### 2.1. Materials

*Bombyx mori* cocoons were kindly supplied by the Langzhong Silkworm Breeding Farm, Sichuan, China. EtOH (Ethanol, purity > 99.5%) was obtained from Chuandong Chemical Co., Ltd., Chongqing, China. LiBr was obtained from Aladdin Chemical Co., Ltd., Shanghai, China. CaCl_2_ was obtained from Kelong Co., Ltd., Chengdu, China. All chemicals used were of analytical grade.

### 2.2. Preparation of Silk Samples

Silk solutions were prepared according to our previously published procedures [21]. Cocoons were cut into small pieces and boiled in 0.02 M Na_2_CO_3_ solution for 30 min, followed by a rinsing process with copious amounts of distilled water to extract sericin proteins. The degummed silk was air-dried at room temperature to obtain the SF. The different fibroin samples were dissolved in a LiBr aqueous solution and a ternary reagent under controlled conditions (Table 1). The resulting solutions were dialyzed in deionized water using a dialysis tube (molecular weight cutoffs of 3500) for 3 days at least, during which time the water was constantly changed. Then, the solution was centrifuged at 5000 rpm for 30 min to remove insoluble impurities. The final concentration was approximately 3–4 wt.% as determined by weighing the remaining solids after drying at 60 °C. The RSF solution was freeze-dried or cast into a thin film to produce the experimental samples. To prepare the silk films, 10 mL of RSF solution was cast on polystyrene dishes (diameter 90 mm) and then dried into films under controlled film-formation conditions (20 ± 5 °C, relative humidity of 65%) for 2 days. In order to unify the standards of films, all silk solutions were diluted to 2 wt.% exactly. 

### 2.3. Molecular Weight Detection

The molecular weight distribution of the RSF was determined with sodium dodecyl sulfate–polyacrylamide gel electrophoresis (SDS–PAGE). Separating gel (15%) and stacking gel (5%) were used in the experiments. An amount of 10 µL of 5× loading buffer was added to 40 µL of RSF solution as the loading sample, 10 µL per well. The electrophoresis voltage of the stacking gel was 80 V, and the voltage of the separating gel was 120 V. After electrophoresis, the samples were stained with Coomassie brilliant blue R250 for 40 min and photographed after destaining.

### 2.4. Morphologic Observation

The morphology of the RSF molecules in solution was observed with an atomic force microscope (AFM; Dimension ICON, Bruker, Germany). Two microliters of the diluted RSF solution (0.1 wt.%) was dropped onto freshly cleaved mica surfaces. The morphology of the RSF in water was observed with an AFM in air. A 225 μm long silicon cantilever, with a spring constant of 3 Nm^−1^ was used in tapping mode with a scan rate of 1 Hz. 

Transmission electron microscopy (TEM; JEM-2100, JEOL, Tokyo, Japan) was also employed to observe the morphology of the RSF molecules. The diluted silk solution (0.1 wt.%) was placed on a carbon-coated Cu electron microscopy grid. The excess liquid was absorbed by filter paper and then air-dried. The sample grid was observed at 80 kV. To enhance the contrast of the image, the sample was dyed with phosphotungstic acid.

### 2.5. Amino Acid Analysis

The amino acid composition of the RSF was determined using an automatic amino acid analyzer (L-8800, Hitachi, Tokyo, Japan). A 50 mg fibroin sample was incubated in 6 M hydrochloric acid at 110 °C for 24 h, then the hydrolyzed solution was transferred to a beaker and evaporated to dryness. The dry mixture was dissolved again with 0.02 mol/L hydrochloric acid and filtered with a 0.22 μm filter. The percent composition of different amino acids was subsequently determined.

### 2.6. Element Content Analysis

Raw SF and the RSF prepared with the two solvents were dried in an oven at 80 °C to a constant weight, and in each case a 0.1 g dry sample was dissolved in 10 mL 65–68% nitric acid at 130 °C, and the solution was evaporated to dryness. The dried mixture was dissolved again with 4 mL of nitric acid and diluted to 50 mL with water. Measurements were performed with an Optima 8000 inductively coupled plasma emission spectrometer (PerkinElmer, Waltham, MA, USA). All measurements were performed in triplicate.

### 2.7. Antibacterial Assays

The microbial growth curve method was employed to evaluate the antibacterial property of the RSF. *Escherichia coli* and *Staphylococcus aureus* were cultured until the mid-log phase (A600 of 0.5–0.6) at 37 °C overnight. A total of 100 μL of lysogeny broth medium was blended with 20 μg fibroin in weight. Each of the blended mixtures was inoculated with bacterial suspensions. The total volume of all groups was 200 μL. All samples were incubated at 37 °C, and the A600 values at different intervals were measured by a Bioscreen C microorganism growth curve meter (Bioscreen, Turku, Finland).

### 2.8. Cell Growth Assessment

The mouse L929 cells used in this study were purchased from Lonza Group Ltd. They were cultured in basal medium supplemented with 10% fetal bovine serum (FBS) in a CO_2_ incubator at 37 °C, according to the published procedure [34]. According to the basic cell culture conditions, 30 μg/mL of RSF solution was added to the basic cell culture medium as the treatment group. Subsequently, a cell counting kit-8 (CCK8) assay was carried out to evaluate L929 cell viability. Using the same cell number and volume, the survival rate was determined by optical density (OD) at 570 nm, with background subtraction at 650 nm, using a Varioskan Flash full wavelength microplate reader (Thermo Scientific, Waltham, MA, USA). Cell images were recorded with an optical microscope.

### 2.9. Hemolysis Test

Whole blood from a rabbit was collected with a vacuum blood collection tube (sodium citrate 1:9), and then diluted with 0.9% sodium chloride for the experiment (0.2 mL in 10 mL saline). The lyophilized RSF was dissolved in 0.9% sodium chloride to prepare three solutions with fibroin concentrations of 100 mg/mL, 10 mg/mL, and 1 mg/mL, respectively, and then centrifuged to remove impurities. An amount of 1 mL of RSF solution was added into a centrifuge tube and incubated in a 37 °C water bath for 30 min. An equal volume of diluted rabbit blood was then added, and incubation continued for 1 h. Physiological saline and distilled water were added as positive and negative controls, respectively. After hemolysis, samples were centrifuged at 12,000 r/min for 2 min. The absorbance of the supernatant was measured at 545 nm using a Synergy H1 microplate reader (BIOTEK, Winooski, VT, USA). The hemolysis rate was calculated according to the following equation:Hemolysis rate (%) = (A2 − A1)/(A3 − A1) × 100
where A1, A2, and A3 are the optical density (OD) of the negative control, sample, and positive control, respectively.

### 2.10. Platelet Adhesion Analysis

The experiment was designed with reference to methods in the published literature [35]. The whole blood was mixed with sodium citrate buffer and centrifuged at 1500 r/min for 15 min to obtain platelet rich plasma (PRP) supernatant. Then it was diluted 1-fold with phosphate buffer saline (PBS, pH7.4), and centrifuged to obtain the supernatant for the experiment. Each RSF film sample was cut into a 1 cm × 1 cm square and placed in a 6-well plate. An amount of 1 mL PBS was added, and it was incubated in a 37 °C water bath for 1 h, and then removed. An amount of 1 mL of PRP diluent was re-added, and then it was placed in an incubator (static test) or a shaker (dynamic test) at 37 °C for 3 h. After the platelets adhered, the RSF film was taken out and gently washed three times with PBS. The film sample was fixed with 2.5% glutaraldehyde solution for 12 h, then dehydrated and dried with different concentrations of ethanol gradient, and then observed with a Crossbeam 350 scanning electron microscope (Zeiss, Jena, Germany).

### 2.11. Film Structural Characterization

Fourier Transform Infrared Spectroscopy (FTIR) analysis of the silk samples was performed with a Nicolet iS5 spectrometer (Thermo Scientific, Waltham, MA, USA) equipped with an attenuated total reflection (ATR) ZnSe crystal. The silk film was attached to the crystal surface with a compressive bar. For each measurement, 32 scans were coded at a resolution of 4 cm^−1^. The wavenumber ranged from 400 to 4000 cm^−1^. The FTIR spectra were fitted with Gaussian profiles in the amide I region between 1600 and 1700 cm^−1^ by PeakFit 4.12 software. Please refer to the Appendix A for the details of the procedures. 

The crystal structure of each film sample was measured with X-ray diffraction (XRD) (X′ Pert3 powder, Malvern Panalytical, Worcestershire, UK) using Cu Kα radiation (20 mA, 36 kV) with a scanning speed of 4°/min. The thermal properties of the silk films were measured in a differential scanning calorimeter (HSC-1, Shanghai, China) under a dry nitrogen gas flow of 10 mL·min^−1^. The samples were heated at 10 °C per min from 25 °C to 350 °C.

The surface morphology of each film was also observed by AFM (Dimension ICON, Bruker, Germany). A small piece of RSF film was placed onto freshly cleaved mica, and its surface morphology was observed in the same way. The operating procedure is as described above.

## 3. Results

### 3.1. Molecular Weight Distribution of the RSF

Native SF is composed of a heavy protein chain (350 kDa) and a light protein chain (26 kDa), which are connected by a disulfide linkage [36]. Since the SF peptide chain is hydrolyzed during the dissolution process, the RSF has a broad molecular weight distribution of 35 kDa to 270 kDa [21,28,29], showing a broad smeared band in the SDS-PAGE gel. From the SDS-PAGE results (Figure 1a), it can be seen that the band distributions of RSF-Li and RSF-Ca are almost the same. This indicates that we successfully prepared two RSFs with the same molecular weight in different solvents by controlling the dissolution conditions. These samples were used for subsequent test comparisons, which effectively avoided the interference of molecular weight on the results and improved the reliability of the test results.

### 3.2. Morphology of the RSF in Solution

In order to compare the molecular-scale morphology of the two SFs, we prepared a fibroin solution at a concentration of 1 mg/mL. Samples were dropped onto a mica surface and observed directly by AFM. The results revealed that the RSF-Li and RSF-Ca molecular structures were uniform and characterized by globular beds of protein (Figure 1b–e). These proteins exhibited the typical long-elliptic morphology, reaching an apparent height of 1.7 nm and a width of 20 nm, approximately. This result was significantly as it corresponded to a previous publication [37]. Using high-resolution AFM, Koebley observed that RSF is granular in aqueous solution [37]. Moreover, TEM images also exhibited the same long-elliptic morphology of the fibrils generated under the same concentration as was seen in the AFM analysis (Figure 1f,g).

### 3.3. Amino Acid Composition of the RSF

The SF consists of 18 amino acids, the most abundant being glycine (G), alanine (A), and serine (S). These three amino acids make up the GAGAGS motif. This motif is repeated in large numbers and constitutes the main structure of SF [38]. It is believed that the composition and sequence of amino acids are closely related to the structure and properties of RSF [39]. Compared with raw SF, the amino acid type in RSF does not change after dissolution and regeneration; only the amino acid content changes slightly [40]. To verify whether the solvent has an effect on the amino acid composition, we tested the amino acid compositions of the two different RSF samples. The results (Figure 2a) showed that the amino acid composition of the SF differed very little before and after dissolution, except for glutamic (Glu) and proline (Pro). The content of glutamic acid in the RSF-Li samples was significantly higher than in the RSF-Ca samples. In addition, the content of proline in the RSF was significantly reduced (Figure 2b). However, the content of these two amino acids in SF is very small, and there is no clear evidence that they can obviously affect the structure of SF.

### 3.4. Content of Ca and Li in the RSF

Natural SF contains trace amounts of Ca, which plays a key role in maintaining the structural and mechanical properties of SF [41,42]. Studies carried out in vitro have also revealed that Ca can induce the formation of β-sheet structures within silk proteins, which improves mechanical properties and promotes gelation in RSF materials [22,43,44,45]. It has been reported that a small amount of Ca is retained in RSF solution which is prepared with calcium salt after dialysis [46]. Therefore, it is necessary to investigate the residual amount of metal elements from the solvent in the two RSF samples prepared using different solvents. Inductively coupled plasma-atomic emission spectroscopy (ICP-AES) was employed for the determination of Ca and Li content, and the results are shown in Figure 3. The Ca content of the native SF (NSF) was 181.7 ± 0.503 mg/kg, but the RSF-Ca had a Ca content of up to 1139.46 ± 3.45 mg/kg, which was about 6 times that of the control. This has demonstrated that a small amount of Ca from the solvent remained in the RSF after dialysis, corresponding to a previously published report [46]. The content of Ca in the RSF-Li was 280.43 ± 3.44 mg/kg, slightly higher than the control, but significantly lower than the RSF-Ca. We speculate that the RSF-Li adsorbs trace amounts of Ca from water during the dialysis process, resulting in a higher Ca content than in NSF (Figure 3a). Li is not present in NSF, so it cannot be detected in the RSF-Ca. However, after three days of dialysis, 117.8 ± 16.89 mg/kg of Li remained in the RSF-Li (Figure 3b). Yet, if the dialysis time is extended (for example, to 7 days), Li will not be detected in the RSF-Li (Figure 3d). This shows that Li can be completely removed from RSF by dialysis. Surprisingly, there will still be a small amount of Ca in the RSF-Ca after 7 days of dialysis (Figure 3c). In addition, the Ca content in the RSF-Li will gradually increase with the extension of dialysis time (Figure 3c). Obviously, the SF molecules can easily and firmly bind to Ca. Previous studies have confirmed that Ca ions can combine with silk protein molecules to form a stable complex structure [46,47,48].

### 3.5. Antibacterial Properties of the RSF

Silk protein exhibits weak antibacterial activity, and the seroin is considered its antibacterial component [49]. To verify whether the solvent has an effect on the antibacterial activity of RSF, we tested the antibacterial activity of the two RSF samples. The results (Figure 4a) show that the RSF has a visible inhibitory effect on *E. coli*, and the two RSF samples exhibit the same antibacterial activity. the survival of *E. coli* in the treatment group was only 75% of that in the control group after 12 h (Figure 4c). However, neither of the two RSF samples inhibited the growth of *S. aureus*, and they even promoted its growth to some extent (Figure 4b). We speculate that the silk protein provides nutrients for the growth of *S. aureus*. These results indicate that under the current experimental conditions, the solvent has no significant effect on the antibacterial properties of RSF.

### 3.6. Cytocompatibility and Hemolysis of the RSF

SF has good biocompatibility and can be used as a medium for animal cell cultures [50]. The two RSF samples were added to a cell culture medium to investigate whether the samples affected cell growth. The results show that the L929 cells grew well in both the media with the silk protein and the control medium. Normal cell morphology was also observed (Figure 5a). More specifically, cellular morphology was normal on the first day of culture. On the third day, the number of cells increased significantly, and their outlines were clear. The cell morphology then changed from fusiform to round, and the cells began to die on the fifth day. No significant differences in cell morphology in terms of cell size, shape, and outline, were observed between the cells cultivated in each medium.

Cell survival was measured using the CCK8 assay. From the results, we found that the L929 cells showed similar overall survival rates in both the fibroin-added media and the control medium, and that the survival rates reached a maximum on the fourth day (Figure 5b). The survival rate of the RSF-added cells, especially the RSF-Ca-added cells, was slightly higher than that of the control cells. In addition, the survival rate of the cells exposed to the RSF-Ca was slightly higher than that of the cells exposed to the RSF-Li. Ca^2+^ is a critical factor for a wide range of physiological processes, and it also plays a regulatory role in cell migration [51]. Therefore, we speculate that the higher Ca content in the RSF-Ca leads to a slightly higher cell survival rate compared with the control. In summary, both RSF-Ca and RSF-Li have no obvious cytotoxicity and have no obvious influence on cellular morphology and cell survival rate.

Previous studies have confirmed that insoluble RSF film does not cause significant hemolysis [34]. Here, we measured the hemolysis rate of the RSF that directly contacted red blood cells in a solution state. It can be seen from the results that the hemolysis rates of the RSF solutions of 1 mg/mL and 10 mg/mL were very low, while the hemolysis rate of the 100 mg/mL RSF solution was high, exceeding 2% (Figure 5c). This shows that high concentrations of fibroin protein are more likely to cause red blood cells to rupture and increase hemolysis when directly contacting red blood cells. However, there is no significant difference in the hemolysis rate between RSF-Li and RSF-Ca, indicating that there is no notable difference in the degree of hemolysis caused by the RSF prepared with either of the two solvents we selected.

### 3.7. Structure of the Silk Film

Thin film is not only an important material form for SF application, but also an important medium for studying the structural transition of fibroin protein. Therefore, we prepared films from two RSF samples and characterized their structures. To ensure the reliability of the results, we strictly controlled the parameters of the film, using the same RSF concentration, temperature, humidity, etc. The FTIR results showed that all samples produced strong absorption bands at 1640 cm^−1^ (amide I) and 1510 cm^−1^ (amide II), and we attribute these bands to random coil and β-sheet conformations, indicating that the two silk films have similar structures (Figure 6a). Further quantitative analysis found that the β-sheet content of the RSF-Ca film was higher than that of the RSF-Li film, while the content of random coils/α-helices was lower (Figure 6b and Appendix A).

The XRD testing showed similar results. All samples exhibited a broad diffraction peak at around 2θ = 22°, indicating the presence of both silk I (type II β-turn) and silk II (anti-parallel β-pleated sheet) structures in the silk film (Figure 6c) [21,52]. The RSF-Li film was a little different, simultaneously showing a typical silk I structure X-ray diffraction peak at 11.8°. This indicates that the silk structure of the RSF-Ca film is mainly composed of stable silk II, which is consistent with the FTIR data. Silk I is a metastable structure, and it is the key intermediate secondary structure formed by silk II, so its thermal degradation temperature is lower than that of the stable silk II structure [52]. The test curve of DSC (Figure 6d) shows that the thermal decomposition temperature of the RSF-Li film is 268 °C, which is slightly lower than that of the RSF-Ca film (284 °C). This is because the RSF-Ca film has a higher β-sheet content, so the thermal stability is higher.

The surface roughness, hydrophilicity, and other micro- and nanostructures of biomaterials are key properties of the materials that can affect various biological properties such as cell adhesion, growth, and aggregation. The surface microstructures of the two silk films were observed using AFM, and the results showed that the two films were not very smooth, but evenly distributed with tiny spherical protrusions (Figure 6e). The roughness analysis results show that the roughness (Ra) of RSF-Ca is slightly higher than that of RSF-Li (Appendix A). Notably, although the secondary structure content, thermal degradation temperature, and surface roughness of the two RSF films were slightly different, the data did not reach a statistically significant level (*p* < 0.05).

As mentioned above, Ca^2+^ can bridge with acidic amino acids in silk protein, thereby promoting the formation of β-sheet structures [22,43,44,45]. Most silks in nature rely on this mechanism, so *Bombyx mori* silk contains trace amounts of Ca to maintain its good mechanical properties [42]. When dissolving SF with calcium salts (e.g., a ternary reagent), after long-term dialysis about 1000 mg/mL of Ca is still bound to the fibroin molecules. Therefore, we speculate that the residual Ca in RSF-Ca forms more β-sheets than in RSF-Li, thereby increasing the thermal degradation temperature and roughness of the film (Figure 6f). The difference was not significant due to the low residual Ca content and the presence of intrinsic Ca in the RSF-Li.

### 3.8. Platelet Adhesion of the RSF Film

Tissue engineering materials are usually not necessary to cause platelet aggregation, especially materials that contact blood directly, such as artificial blood vessels. In addition, Ca^2+^ is a procoagulant factor that can cause platelet aggregation to stop bleeding. There are trace Ca residues in RSF-Ca, so it is necessary to evaluate its platelet adhesion performance. In this experiment, the adhesion of rabbit platelets to RSF films was evaluated. The experimental results showed that platelets did adhere more to the RSF film under static conditions, but that they did not adhere well under dynamic conditions that mimicked blood flow (Figure 7a and Appendix A). In addition, we conducted a statistical analysis of the number of platelets per unit film area. It was found that the number of platelets was higher on the RSF-Ca film than on the RSF-Li film, but the difference was not significant (Figure 7b). This shows that these two kinds of RSF are safe and can be used to prepare biological materials that contact with blood directly.

## 4. Conclusions

The effect of a ternary reagent and of LiBr on the structure and properties of RSF were systematically investigated in the present study. The results demonstrated that RSF prepared using a ternary reagent was not significantly different from that prepared using LiBr in terms of amino acid composition, morphology, bacteriostatic activity, cytotoxicity, film structure, and properties. Therefore, we believe that a ternary reagent is an ideal alternative solvent for dissolving SF. Notably, compared with LiBr, trace amounts of Ca remained in the RSF dissolved by the ternary solvent, which is hard to remove completely, even after long-term dialysis. This resulted in subtle changes in the structure and properties of the material. The study also reminds us that the electronegativity of SF molecules and specific amino acid residues (e.g., Asp and Glu) make it relatively easy to bind it to metal elements such as Ca, Mg, Fe, and Zn, so its solubility and preparation process should be considered carefully.

## Figures and Tables

**Figure 1 polymers-14-03903-f001:**
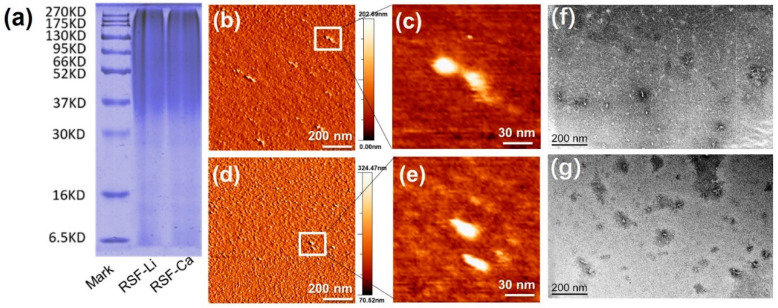
Molecular weight and morphology of the RSF. (**a**) The SDS-PAGE result. (**b**–**e**) The AFM images of the RSF-Ca (**b**,**c**) and the RSF-Li (**d**,**e**) in solution. (**f**,**g**) The TEM images of the RSF-Ca (**f**) and the RSF-Li (**g**). The arrows (**e**,**f**) indicate the RSF.

**Figure 2 polymers-14-03903-f002:**
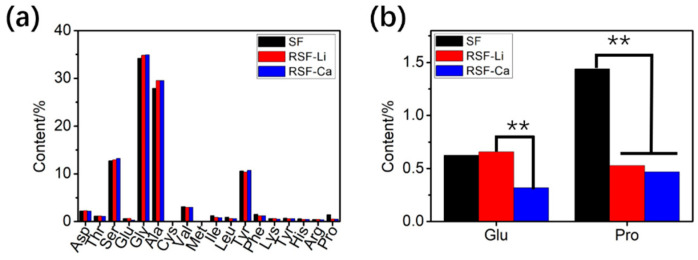
The amino acid composition of silk protein. (**a**) The content of 18 amino acids. (**b**) The content of Glu and Pro was significantly different during the dissolution process. (** *p* < 0.01).

**Figure 3 polymers-14-03903-f003:**
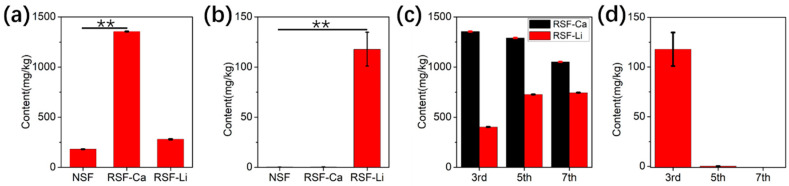
The elemental content in the RSF. (**a**) The content of Ca. (**b**) The content of Li. (**c**) The Ca content in the two RSF samples on days 3, 5 and 7 of dialysis. (**d**) The Li content in the RSF-Li on days 3, 5 and 7 of dialysis. Prolonging the dialysis time can completely remove Li, but not Ca. (** *p* < 0.01, *n* = 3).

**Figure 4 polymers-14-03903-f004:**
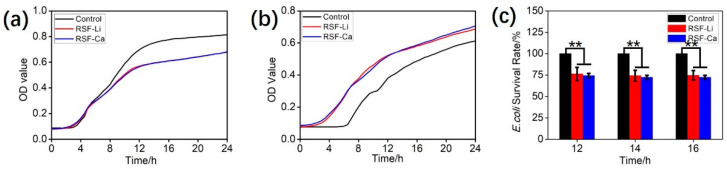
Antibacterial properties of the RSF-Li and RSF-Ca. Growth curves of (**a**) *E. coli* and (**b**) *S. aureus*. (**c**) *E. coli* survival upon incubation with RSF at the following time points: 12, 14 and 16 h (** *p* < 0.01, *n* = 5).

**Figure 5 polymers-14-03903-f005:**
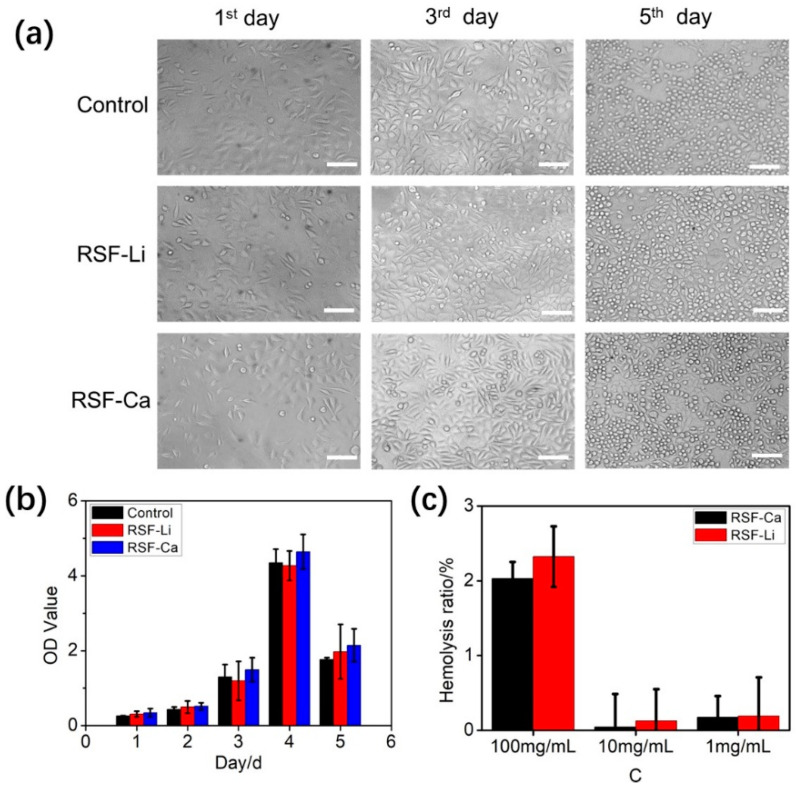
The cytocompatibility and hemolysis of the RSF. (**a**) The cellular morphology and (**b**) cell survival rate of mouse L929 cells in the control medium and the RSF-added media. Cellular morphology was observed on the first day, the third day, and the fifth day. (**c**) The hemolysis rate of the RSF that directly contacted red blood cells in a solution state. The scale bars in (**a**) represent 100 μm.

**Figure 6 polymers-14-03903-f006:**
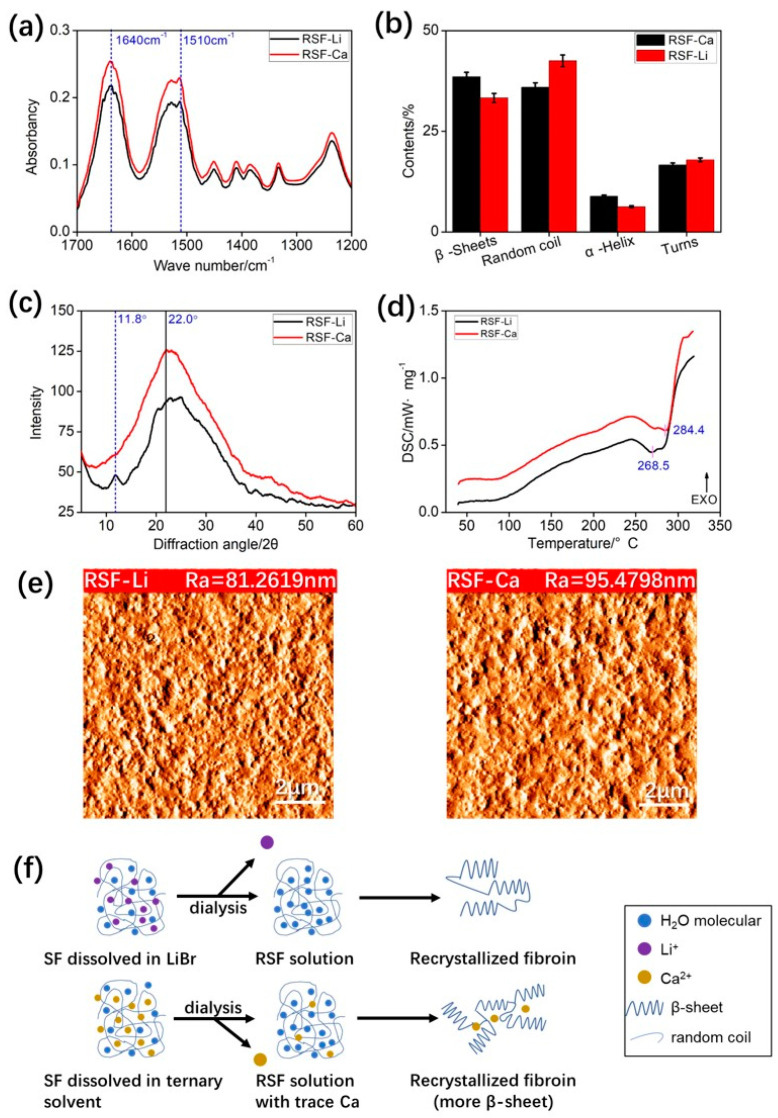
The structural analysis of the RSF films. (**a**) The FTIR spectra and (**b**) the secondary structure of the two RSF films. (**c**) The XRD and (**d**) the DSC curves of the two RSF films. (**e**) The AFM photos of the two RSF films. (**f**) A schematic illustration of the effect of residual Ca in RSF on the self-assembly of fibroin molecules.

**Figure 7 polymers-14-03903-f007:**
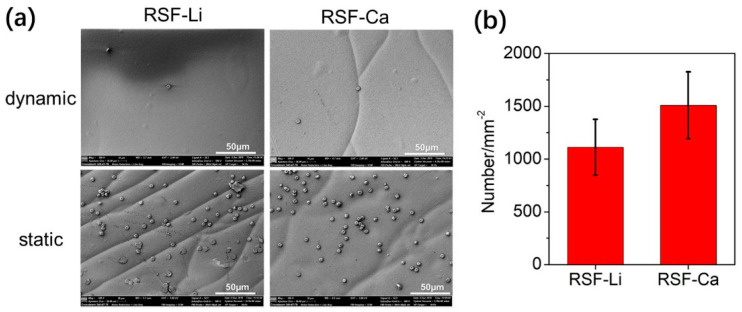
The platelet adhesion of the two RSF samples. (**a**) The morphology of the platelets adhered to the RSF film. (**b**) Platelet adhesion per unit area.

**Table 1 polymers-14-03903-t001:** Preparation of RSF under controlled conditions.

Solvent	Preparation	Temperature (°C)	Time (min)	Bath Ratio
LiBr	9 M LiBr aqueous solution	80	3	1:10
Ternary reagent	CaCl_2_–EtOH–H_2_O with a molar ratio of 1:2:8	75	15	1:10

## Data Availability

The data presented in this study are available on request from the corresponding author.

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
