# Peer review of "Insignificant Difference in Biocompatibility of Regenerated Silk Fibroin Prepared with Ternary Reagent Compared with Regenerated Silk Fibroin Prepared with Lithium Bromide"

_polymers, 2022, doi:10.3390/polym14183903_

Round 1

Reviewer 1 Report

This study describes regenerated silk fibroin prepared with ternary reagent shows an insignificant difference compared to lithium bromide in biocompatibility. The work lacks novelty since a previous report is HIGHLY SIMILAR to the present study: Shen, Tingting, Tao Wang, Guotao Cheng, Lan Huang, Lei Chen, and Dayang Wu. "Dissolution behavior of silk fibroin in a low concentration CaCl2-methanol solvent: From morphology to nanostructure." International journal of biological macromolecules 113 (2018): 458-463.

Author Response

Response to Reviewer 1 Comments

Point 1: This study describes regenerated silk fibroin prepared with ternary reagent shows an insignificant difference compared to lithium bromide in biocompatibility. The work lacks novelty since a previous report is HIGHLY SIMILAR to the present study: Shen, Tingting, Tao Wang, Guotao Cheng, Lan Huang, Lei Chen, and Dayang Wu. "Dissolution behavior of silk fibroin in a low concentration CaCl2-methanol solvent: From morphology to nanostructure." International journal of biological macromolecules 113 (2018): 458-463.

Response 1: Thanks for you question. The article you mentioned mainly reported the solubilization behavior of silk fibroin in CaCl2--methanol solvent. Its innovation and focus lies in the first report of a new solvent that can dissolve silk fibroin. This low concentration of CaCl2-methanol solvent can degrade silk fibroin into nanofibers, but not completely dissolve it. Therefore, it is not equivalent to CaCl2-ethanol-water ternary solvent. Our study focuses on the large-scale and low-cost preparation of regenerated silk fibroin, and systematically compares the difference between regenerated silk fibroin prepared by inexpensive ternary solvent and expensive LiBr. This is of great significance in the current era of rapid development of new energy and soaring prices of lithium salts. Additional, I'm one of the authors of that 2018 article.

Reviewer 2 Report

1. Need grammar checking, more literature review, and point of view of this research

2. line 14-15: revised to:Recently,  lithium bromind (LiBr) is a traditional solvent which is usually used to dissolve SE (give more detail to explain the properties of LiBr). However, LiBr has several some limitations including expensive, organism toxic, and environmental friendly. Herein, ternary reagent system is proposed to be studied under the condition of mixture CaCl2, EtOH, and H2O with molar ration :1:2:8.

3. Following above mention, the mixture CaCl2, EtOH, and H2O with molar ration :1:2:8 was proposed as the optimal condition so that please explain.

4. I am not sure the meaning of RSF. RSF: Regenerated silk fibroin?

5. line 40: More efficient new solvents: Please give examples of solvents and how they make them more efficient compared to conventional solvents.

6. Why RSF was proposed to be studied? please give the strongly main idea of your research.

7.  To prepare the RSF, please give more detail about the advantages of ternary reagent system for using as a mediated-solvent system.

8. Through the preparation of the RSF via ternary reagent system how this synthesis route can overcome the limitation of LiBr.

9. line 53: revised report to reported

10. Under ternary reagent, what is the superiors properties of RSF generated under studied condition than LiBr.

11. line 208: revised to: this result was significantly corresponded to a previous publication (so please give more detail of the result form the a previous publication).

Moreover, TEM  images also exhibited the  long-elliptic morphology of fibril generated under the same concentration which was attributed to AFM analysis.

12. line 215: please explain more detail of amino acid composition of RSF and provide the main function of amino acid in RSF.

13. line 219: the slightly changes in amino acid content, are their exhibiting any effects on the RSF?

14. line 241:With the increasing Ca content, it could be demonstrated that Ca ions could combine with silk protein molecules to form a stable complex structure.

So, how about if Ca content was increased six times? Is it optimized performance, please explain?

15. Following the mention above, please explain the mechanism of structure stabilization by combining Ca ions and silk protein molecules.

Author Response

Response to Reviewer 2 Comments

  1. Need grammar checking, more literature review, and point of view of this research

Response 1: Thanks for your advice. Our manuscripts have been previously polished by professional companies (LetPub). We have re-checked the manuscript and corrected some grammatical errors and spelling errors. Based on your comments and those of other reviewers, we read more literature and referenced some relevant literature. And some points of the article have been revised and improved.

  1. line 14-15: revised to: Recently,  lithium bromind (LiBr) is a traditional solvent which is usually used to dissolve SE (give more detail to explain the properties of LiBr). However, LiBr has several some limitations including expensive, organism toxic, and environmental friendly. Herein, ternary reagent system is proposed to be studied under the condition of mixture CaCl2, EtOH, and H2O with molar ration :1:2:8.

Response 2: Thanks for your suggestion. Lithium bromide (Libr) is not an environmentally friendly solvent and is highly toxic to the ecological environment. In the context of the vigorous development of new energy, the price of lithium salt has soared to an unbearable level. Combined with processing and recycling costs, the production of silk fibroin using lithium bromide is almost impossible to industrialize. It is not the first time that ternary solvents have been proposed and used, and our research focuses on analyzing whether regenerated silk fibroin produced by ternary solvents can match or outperform the silk fibroin produced by lithium bromide. This provides a theoretical basis for the substitution of lithium bromide. So, based on your comment, I am modifying this paragraph to:  

> Lithium bromide (LiBr) is a traditional solvent which is usually used to dissolve SF. However, LiBr has several limitations including expensive, organism toxic, and environmental unfriendly. Herein, a ternary reagent system that is inexpensive, organism non-toxic, and environmental friendly is proposed to be investigated as an alternative to silk fibroin solubilization.

  1. Following above mention, the mixture CaCl2, EtOH, and H2O with molar ration :1:2:8 was proposed as the optimal condition so that please explain.

Response 3: Thanks for your question. Ternary solvents were first proposed by Japanese scholars in 1997. In the ratio of 1:2:8, the rapid dissolution of silk fibroin can be achieved. Details can be found in the literature below.

>Ajisawa A. Dissolution aqueous of silk fibroin with calcium chloride/ethanol solution. J Seric Sci Jpn. 1997;67:91–94.

  1. I am not sure the meaning of RSF. RSF: Regenerated silk fibroin?

Response 4: I'm very sorry for the inconvenience caused to you. RSF is the abbreviation of regenerated silk fibroin, which refers to the silk fibroin obtained after solvent dissolution and dialysis. I have explained it in detail when it first appeared in the article so as not to cause unnecessary confusion.

  1. line 40: More efficient new solvents: Please give examples of solvents and how they make them more efficient compared to conventional solvents.

Response 5: Thanks for your review. My original intention: more new solvents that can efficiently dissolve silk fibroin. This misstatement has been revised in the article.

  1. Why RSF was proposed to be studied? please give the strongly main idea of your research.

Response 6: Thanks for your question. Regarding the solubilization of silk fibroin, the solvent is an important research field. Our work mainly focuses on the solubilization of silk fibroin, especially the effect of different solvents. Several previous reports have compared in detail the differences in yield, solubilization ability, and solubilization conditions of different solvents in the process of dissolving silk fibroin, including the work of our team. LiBr and ternary solvent are very similar in terms of solubility and yield, and both can rapidly and completely dissolve silk fibroin under conventional conditions. Laboratories and teams around the world studying silk fibroin materials are basically divided into two groups, one using LiBr and the other using ternary solvents. No researchers could definitively tell the difference between the two solvents, especially how it affects the final product--RSF. Our research is to answer this question thoroughly and provide a reference for the low-cost and industrialized production of regenerated silk fibroin.

  1. To prepare the RSF, please give more detail about the advantages of ternary reagent system for using as a mediated-solvent system.

Response 7: Thanks for your suggetion. The advantages of ternary solvents are mainly reflected in the following aspects: low price of raw materials, harmless to organisms and the environment, and low cost of wastewater treatment. If the silk fibroin produced is no different from lithium bromide or slightly better than lithium bromide, it will be very advantageous in industrial production. Following your advice, I have intensified the discussion of the advantages of ternary solvents in the Discussion section.

  1. Through the preparation of the RSF via ternary reagent system how this synthesis route can overcome the limitation of LiBr.

Response 8: Thanks for your question. The limitation of LiBr lies mainly in itself and has nothing to do with the synthetic route. The same is true for the advantages of ternary solvents.

  1. line 53: revised report to reported

Response 9: Thanks for your suggetion. This error has been corrected in the manuscript.

  1. Under ternary reagent, what is the superiors properties of RSF generated under studied condition than LiBr.

Response 10: Thanks for your quesrion. Our investigation found that trace amounts of Ca remained in the RSF prepared by ternary solvent. Ca is an essential element in the composition of life. Trace amounts of Ca present in RSF can increase cell viability when cells are co-cultured with RSF. In addition, low concentrations of Ca can promote the structural transition of silk fibroin to β-sheet. More β-sheets mean more stable structures. Therefore, the silk fibroin material prepared by ternary solvent has better mechanical properties. It should be noted that the residual amount of Ca is very low, so the above performance improvement is not significant compared with the RSF prepared by LiBr. Therefore, we believe that the most advantage of ternary solvents is still the low production cost.

  1. line 208: revised to: this result was significantly corresponded to a previous publication (so please give more detail of the result form the a previous publication).

Moreover, TEM  images also exhibited the  long-elliptic morphology of fibril generated under the same concentration which was attributed to AFM analysis.

Response 11: Thanks for your suggetion. Following your suggestion, I have revised this in the manuscript.

  1. line 215: please explain more detail of amino acid composition of RSF and provide the main function of amino acid in RSF.

Response 12: Thanks for your suggetion. The amino acid composition and function of SF has been added to the manuscript.

  1. line 219: the slightly changes in amino acid content, are their exhibiting any effects on the RSF?

Response 13: Thanks for your question. The properties of native SF and RSF are significantly different. Natural SF has high crystallinity and strong mechanical properties. Current studies tend to believe that the structure of native SF is severely damaged during the dissolution process, thereby affecting the structure and mechanical properties of RSF. There is no clear evidence that it is caused by amino acid changes, but it cannot be ruled out that the destruction of some amino acids during the dissolution process leads to changes in the structure and properties of native SF and RSF.

  1. line 241: With the increasing Ca content, it could be demonstrated that Ca ions could combine with silk protein molecules to form a stable complex structure.

So, how about if Ca content was increased six times? Is it optimized performance, please explain?

 Response 14: Thanks for your question. In our study, the content of Ca in RSF prepared by ternary solvent was 6 times that of native SF and 4 times that of RSF prepared by LiBrary. And after 7 days of dialysis, the Ca content did not decrease. We deduce that this is because calcium ions bind stably with certain amino acids of the silk protein molecule. From the results of subsequent test analysis, the RSF film prepared by ternary solvent is slightly better than LiBr.

15.Following the mention above, please explain the mechanism of structure stabilization by combining Ca ions and silk protein molecules.

Response 15: Thanks for your question. There is evidence that Ca ions can bridge between acidic amino acids (Asp and Glu). Ca ions are bridged with acidic amino acids to form a stable complex, which causes a slight twist of the silk fibroin molecular chain, thereby forming a stable folded structure. I have added a description of this mechanism at the appropriate place in the manuscript and cited the latest findings.

Reviewer 3 Report

Cheng et al. make an important point regarding LiBr, which is commonly used as the basis for redissolving silk fibroin.  Not only is the salt potentially harmful to humans and the environment, it is expensive - and likely to become more expensive - due to competition from the energy industry for battery construction.

The authors present an interesting and wide-ranging comparison between fibroin redissolved using concentrated aqueous LiBr or the ternary (CaCl2/EtOH/water) system.  In my opinion, the manuscript is well written and I have only a few minor comments:

1) Line 34: The authors state that fibroin is water-insoluble, but that is something of an oversimplification and requires clarification.  Actually, native silk fibroin protein inside the silkworm is water-soluble and behaves like a typical soluble polymer in solution, e.g: see NMR studies by Asakura and coworkers or rheology studies by Laity et al.

> Asakura, T. Okushita, K. Williamson, MP. Analysis of the Structure of Bombyx mori Silk Fibroin by NMR,  Macromolecules 2015, 48, 2345−2357, https://doi.org/10.1021/acs.macromol.5b00160

> Asakura, T. Structure of Silk I (Bombyx mori Silk Fibroin before Spinning) -Type II beta-Turn, Not alpha-Helix, Molecules 2021, 26, 3706 https://doi.org/10.3390/molecules26123706

> Laity, PR. Holland, C. Thermo-rheological behaviour of native silk feedstocks, Eur. Polym.  J. 87 (2017) 519–534 http://dx.doi.org/10.1016/j.eurpolymj.2016.10.054

> Laity, PR. Gilks, SE. Holland, C. Rheological behaviour of native silk feedstocks, Polymer 67 (2015) 28e39 http://dx.doi.org/10.1016/j.polymer.2015.04.049

Indeed, if fibroin were not water-soluble, dialysis after dissolution in either solvent system (LiBr or CaCl2-based) would cause immediate precipitation.  Clearly, once it is spun, however, the silk fibre will not dissolve in water.  The important deciding factors may be related to hydration of the native protein and formation of ß-sheets in the spun fibre, e.g. see:

> Laity, PR. Holland, C. Seeking Solvation: Exploring the Role of Protein Hydration in Silk Gelation, Molecules 2022, 27, 551, https://doi.org/10.3390/molecules27020551

I suggest the authors should discuss this point in more detail, please.

2) Page 1: In order to avoid confusion, please define the abbreviations SF and RSF in the main text.  Is SF silk fibroin or silk fibre?  Is RSF redissolved (or reconstituted) silk feedstock or redissolved silk fibroin?  Clearly, fibre indicates a solid, silk feedstock implies a liquid, while fibroin could be either solid or in solution.

3) In several places, the experimental methods were described as instructions, in the imperative.  Eg.

L99: 'Add 10 µL of 5x loading buffer ...'
L125: '...and make up to...'
L151-153: 'Take 1 mL of fibroin solution into a centrifuge tube, and keep it in a 37°C water bath for 30 minutes. Then add an equal volume of diluted rabbit blood, and continue to incubate for 1 hour.'

Please, convert all the various descriptions to the past passive tense to match the rest of the methodology.

4) Lines 130 and 140: Please specify if the RSF and fibroin were solids or in solution.

5) In Figure 1 (b-e), please provide colour scales for the vertical heights of the AFM images.

6) Regarding changes in amino acid concentration and residual ions, there is evidence that Ca can bridge between acidic amino acids (Asp and Glu), which can affect the rheology of fibroin solutions.  See:

> Laity, PR. Baldwin, E. Holland, C.  Changes in Silk Feedstock Rheology during Cocoon Construction: The Role of Calcium and Potassium Ions, Macromol. Biosci. 2018, 1800188, DOI: 10.1002/mabi.201800188

> Schaefer C. et al. Silk Protein Solution: A Natural Example of Sticky Reptation, Macromolecules 2020, 53, 2669−2676, https://dx.doi.org/10.1021/acs.macromol.9b02630

Although the authors show data (Fig. 1a) suggesting similar molecular weight distributions for their RSF-Ca and RSF-Li materials, recent research has suggested that Ca bridges between acidic groups on the fibroin can raise the viscosity.  Consequently, I suggest it would be interesting to measure the rheology of the two RSF solutions.

7) Regarding antibacterial activity of the RSF, would the seroins be extracted with sericin, during degumming?  Can the authors comment, please?

There was also a minor typographical error:

L91: The phrase should be '...or cast into a thin film...'

Author Response

Response to Reviewer 3 Comments

Point 1: Cheng et al. make an important point regarding LiBr, which is commonly used as the basis for redissolving silk fibroin.  Not only is the salt potentially harmful to humans and the environment, it is expensive - and likely to become more expensive - due to competition from the energy industry for battery construction.

Response 1: Thanks for your affirmation.

Point 2:The authors present an interesting and wide-ranging comparison between fibroin redissolved using concentrated aqueous LiBr or the ternary (CaCl2/EtOH/water) system.  In my opinion, the manuscript is well written and I have only a few minor comments:

1) Line 34: The authors state that fibroin is water-insoluble, but that is something of an oversimplification and requires clarification.  Actually, native silk fibroin protein inside the silkworm is water-soluble and behaves like a typical soluble polymer in solution, e.g: see NMR studies by Asakura and coworkers or rheology studies by Laity et al.

> Asakura, T. Okushita, K. Williamson, MP. Analysis of the Structure of Bombyx mori Silk Fibroin by NMR,  Macromolecules 2015, 48, 2345−2357, https://doi.org/10.1021/acs.macromol.5b00160

> Asakura, T. Structure of Silk I (Bombyx mori Silk Fibroin before Spinning) -Type II beta-Turn, Not alpha-Helix, Molecules 2021, 26, 3706 https://doi.org/10.3390/molecules26123706

> Laity, PR. Holland, C. Thermo-rheological behaviour of native silk feedstocks, Eur. Polym.  J. 87 (2017) 519–534 http://dx.doi.org/10.1016/j.eurpolymj.2016.10.054

> Laity, PR. Gilks, SE. Holland, C. Rheological behaviour of native silk feedstocks, Polymer 67 (2015) 28e39 http://dx.doi.org/10.1016/j.polymer.2015.04.049

Indeed, if fibroin were not water-soluble, dialysis after dissolution in either solvent system (LiBr or CaCl2-based) would cause immediate precipitation.  Clearly, once it is spun, however, the silk fibre will not dissolve in water.  The important deciding factors may be related to hydration of the native protein and formation of ß-sheets in the spun fibre, e.g. see:

> Laity, PR. Holland, C. Seeking Solvation: Exploring the Role of Protein Hydration in Silk Gelation, Molecules 2022, 27, 551, https://doi.org/10.3390/molecules27020551

I suggest the authors should discuss this point in more detail, please.
Response 2: Thanks for your suggestion. I have modified this in the article (in red) and discussed the solubility changes of silk fibroin in Bombyx mori and after spinning, and cited the above literature.

Point 3: Page 1: In order to avoid confusion, please define the abbreviations SF and RSF in the main text.  Is SF silk fibroin or silk fibre?  Is RSF redissolved (or reconstituted) silk feedstock or redissolved silk fibroin?  Clearly, fibre indicates a solid, silk feedstock implies a liquid, while fibroin could be either solid or in solution.
Response 3: Thanks for your question. The SF is an abbreviation for silk fibroin, which refers to the insoluble silk fibroin obtained after degumming of cocoons. The RSF is an abbreviation for regenerated silk fibroin, and refers to soluble silk fibroin obtained by dissolving insoluble silk fibroin. Full names and their abbreviations are clearly defined on their first occurrence in the article.

Point 4: In several places, the experimental methods were described as instructions, in the imperative.  Eg.

L99: 'Add 10 µL of 5x loading buffer ...'
L125: '...and make up to...'
L151-153: 'Take 1 mL of fibroin solution into a centrifuge tube, and keep it in a 37°C water bath for 30 minutes. Then add an equal volume of diluted rabbit blood, and continue to incubate for 1 hour.'

Please, convert all the various descriptions to the past passive tense to match the rest of the methodology.
Response 4: Thanks for your suggestion. The question has been revised in the article. Furthermore, I double-checked all statements to make sure there was no identical mistakes.

Point 5: Lines 130 and 140: Please specify if the RSF and fibroin were solids or in solution.
Response 5: Thanks for your suggestion. In the experiment, lyophilized silk fibroin was prepared into a certain concentration solution and used. Line 130 is also the solution used and this expression represents the concentration of silk fibroin. To avoid confusion, I have made clarifications and revisions in the text.

Point 6: In Figure 1 (b-e), please provide colour scales for the vertical heights of the AFM images.
Response 6: Thank you. The picture has been modified in the article. It should be noted that the previous photos were made using the analysis software of the testing instrument itself. I couldn't run the software on my computer, so I recreated the AFM pictures from the original data. Due to software limitations, colour scales cannot be added to the enlarged images of c and e.

Point 7: Regarding changes in amino acid concentration and residual ions, there is evidence that Ca can bridge between acidic amino acids (Asp and Glu), which can affect the rheology of fibroin solutions.  See:

> Laity, PR. Baldwin, E. Holland, C.  Changes in Silk Feedstock Rheology during Cocoon Construction: The Role of Calcium and Potassium Ions, Macromol. Biosci. 2018, 1800188, DOI: 10.1002/mabi.201800188

> Schaefer C. et al. Silk Protein Solution: A Natural Example of Sticky Reptation, Macromolecules 2020, 53, 2669−2676, https://dx.doi.org/10.1021/acs.macromol.9b02630

Although the authors show data (Fig. 1a) suggesting similar molecular weight distributions for their RSF-Ca and RSF-Li materials, recent research has suggested that Ca bridges between acidic groups on the fibroin can raise the viscosity.  Consequently, I suggest it would be interesting to measure the rheology of the two RSF solutions.
Response 7: Thanks for your advice. We noticed this phenomenon early on and tested the viscosity of regenerated silk fibroin solutions prepared with lithium bromide and a ternary solvent (Fig. 1). I did not discuss this issue in depth in my article, as there was research as early as 2001 (Ref. 9) that explored it in depth. In addition, we found through experiments that the viscosity of the two silk fibroin solutions differed significantly only at higher concentrations. This indicated that residual trace Ca had little effect on the regenerated silk fibroin prepared by ternary solvent. Therefore, it should not be overemphasized so as not to conflict with the theme.

Fig. 1 Viscosity of regenerated silk fibroin prepared by ternary solvent and lithium bromide at different concentrations. A: 5%; B: 10%; C: 15%.

Point 8: Regarding antibacterial activity of the RSF, would the seroins be extracted with sericin, during degumming?  Can the authors comment, please?
Response 8: Thanks for your question. Several studies have shown that the seroin is one of the main antimicrobial proteins in silkworm cocoons. The seroin can be synthesized and secreted in multiple parts of silk glands of Bombyx mori, including the posterior silk gland. Therefore, the silk fibroin contains seroin proteins, and degumming does not completely remove them. The seroin can also be detected in degummed silk fibroin. Details can be found in the following literature.

> Dong, Z. et al. Analysis of proteome dynamics inside the silk gland lumen of Bombyx mori. Sci. Rep. 6, 21158; doi: 10.1038/srep21158 (2016)

> Zhang Y.et al. Protein Components of Degumming Bombyx mori Silk. Scientia Agricultura Sinica, 2018. doi: 10.3864/j.issn.0578-1752.2018.11.018

Point 9:There was also a minor typographical error:
L91: The phrase should be '...or cast into a thin film...'

Response 9: Thanks for your meticulous work. Errors have been corrected in the text.

Round 2

Reviewer 1 Report

There are many published papers with very similar contents to the present work. For example: Zhang, Meng, Yu‐Jie Weng, and Yu‐Qing Zhang. "Accelerated desalting and purification of silk fibroin in a CaCl2‐EtOH‐H2O ternary system by excess isopropanol extraction." Journal of Chemical Technology & Biotechnology 96, no. 5 (2021): 1176-1186.

Miyaguchi, Yuji, and Jianen Hu. "Physicochemical properties of silk fibroin after solubilization using calcium chloride with or without ethanol." Food Science and Technology Research 11, no. 1 (2005): 37-42.

Author Response

Point 1: There are many published papers with very similar contents to the present work. For example: Zhang, Meng, Yu‐Jie Weng, and Yu‐Qing Zhang. "Accelerated desalting and purification of silk fibroin in a CaCl2‐EtOH‐H2O ternary system by excess isopropanol extraction." Journal of Chemical Technology & Biotechnology 96, no. 5 (2021): 1176-1186.

Miyaguchi, Yuji, and Jianen Hu. "Physicochemical properties of silk fibroin after solubilization using calcium chloride with or without ethanol." Food Science and Technology Research 11, no. 1 (2005): 37-42.

Response 1: Thanks for your suggestion. ​Following your suggestion, we cite relevant research extensively and expand the research background in the INTRODUCTION, especially the novelty of the present work and the difference from the previous work. A statement and research progress on ternary solvents have been added to the manuscript. In addition, the role of ethanol in dissolving silk fibroin was also added.
